# Establishment of methanogen bacterial interactions during the preweaning period of dairy cattle

**Nagaraju Indugu, Kapil S. Narayan, Meagan L. Hennessy, Dipti Pitta** *

Department of Clinical Studies, School of Veterinary Medicine, University of Pennsylvania, Kennett Square, PA, United States of America

* dpitta@vet.upenn.edu

**Data Availability Statement:** The fecal archaeal 16s rRNA sequences have been deposited in the NCBI database under BioProject accession number PRJNA1046510.

## Abstract

Ruminant livestock are major contributors to anthropogenic methane emissions in the United States and worldwide. Enteric methane is generated by methanogenic archaea residing in ruminant digestive tracts. Information on when methanogens colonize the gut and when they begin to interact with bacteria during the early phases of the ruminant life cycle is less explored. The objectives of this study were (i) to investigate the composition of the methanogenic archaeal community at birth and through the weaning transition and (ii) to determine if and when the methanogenic archaea begin to interact with bacteria in the lower gut of neonatal dairy calves. Ten female Holstein calves (approximately 45kg birth weight) were enrolled in the study. Fecal samples were collected every two weeks (Wk 2, 4, 6, 8, 10, and 12) between birth and weaning and analyzed for methanogenic archaeal diversity via 16S rRNA amplicon sequencing and quantitative real-time PCR (RT-qPCR). Estimates of alpha diversity (Observed species, and Shannon diversity index) and beta diversity (weighted and unweighted UniFrac distances) showed significant differences ($P < 0.05$) between archaeal communities across timepoints. Both 16S rRNA amplicon sequencing and RT-qPCR analyses revealed *Methanobrevibacter* was the most prevalent genus at Wk2, Wk4, and Wk6, whereas *Methanosphaera* gradually increased with time and was most abundant at Wk10 and Wk12. Correlation analysis revealed that *Methanobrevibacter* and *Methanosphaera* were inversely correlated with each other and formed distinct cohorts with specific bacterial lineages similar to those reported in the mature rumen, thus revealing that these associations are established during the preweaning period. Therefore, the preweaning period presents a window of opportunity to interfere with early-life methanogenic colonization with the ultimate goal of reducing enteric methane emissions without perturbing ruminal function later in the life of dairy cattle.

## Introduction

Methane, the second most prevalent anthropogenic greenhouse gas (GHG) after carbon dioxide ($CO_2$), contributes approximately 20% to global emissions and possesses over 25 times the

**Funding:** This research was funded by the United States Department of Agriculture (Grant #PENV570262).

**Competing interests:** The authors have declared that no competing interests exist.

heat-trapping potency of $CO_2$. In the United States, ruminants play a substantial role in the methane footprint, with 27% originating from enteric methane and 9% from manure [1]. Enteric methane emissions from farmed ruminants are considered a major problem not only for the environment but also an energy loss to the ruminant host. An adult ruminant host can produce between 250 and 500 L of methane per day [2]. Enteric methane is exclusively produced by methanogenic archaea (a special group of microbes that exist in synergy with other fermenting microbes) during microbial fermentation of feeds in the rumen [3]. While it is known that methanogens are important for the adult rumen to keep the $H_2$ concentrations low for fermentation reactions to occur, information on their colonization, population dynamics, and the evolution of their interactions with other microbes during the preweaning stage, and as they transition to mature ruminants is not very well explored.

This study posits the question, are methanogens present in the gut at birth? Studies have shown that methanogens occur in the rumen as early as 20 min after birth [4] and during the first week of life [5, 6]. A study by Friedman et al., [7] investigated the copy number of total methanogens (16S rRNA gene) and individual methanogens in the rumen fluid of newborn (2–3 days old) dairy calves. The authors reported that the copy number of methanogenic populations was lower in the first 2-3d, and no methane was emitted by the newborn calves. However, there was a sharp increase of 1 x $10^5$ copies of methanogens/10ng of DNA with about 1100 nmol/mL of methane production that was measured *in vitro*. Notably, the order *Methanobacteriales* was found higher (1 x $10^3$) in copy numbers/10ng of DNA compared to *Methanomicrobiales*, *Methanosarcinales* and *Methanoplasmatales*. Further, these authors also stated that methylotrophic methanogens (The methanogens that produce methane by utilizing fermentation products such as methylamines and methanol) were most predominant in the early stages of life, which are then replaced with hydrogenotrophic methanogens (The methanogens that produce methane by using hydrogen gas ($H_2$) and carbon dioxide ($CO_2$) as substrates) in mature dairy cattle. Another study by Zhou et al., [8] investigated methanogen distribution along the gastrointestinal tract of calves aged 3–4 wks and reported that methanogens closely resembled *Methanobrevibacter* with *M. 1* and *smithii*. Although lineages resembling *Methanosphaera* were detected, their abundance was variable among animals. The total abundance of methanogens varied from $7.9 \times 10^5$ to $3.9 \times 10^8$ copies/gram of rumen content, with *Methanobrevibacter sp.* and *M. stadtmanae* contributing up to 5.2 x $10^7$ copies up to 2.8 x $10^8$ copies/gram of rumen content, respectively. While previous studies have focused on the early methanogens colonizers in adult ruminants at any single timepoint [4, 9], there is a noticeable knowledge gap regarding their establishment in neonatal and growth phases of ruminant host's life cycle. Particularly, when do methanogens colonize the gut, their establishment, the diversity and dynamics of archaeal populations in healthy calves from birth, through weaning and as the calves transition to a solid diet.

This study aims to enhance our understanding of the changes in calf fecal methanogen populations and how they interact with bacteria from shortly after birth though 12 weeks of age. We hypothesized that significant fluctuations occur in fecal archaeal communities in response to changing dietary regimen during the neonatal period of dairy calves. These changes in methanogen communities will further bacterial community composition via their partnerships with specific gut bacteria.

## Materials and methods

### Study design and sample collection

This study complements prior research on bacterial communities conducted by Hennessy et al., [10] by focusing on the temporal dynamics of archaeal communities in 10 female

Holstein dairy calves. Ethical approval for all procedures involving animal subjects was obtained from the University of Pennsylvania Institutional Animal Care and Use Committee (IUCAC protocol #80614). The cohort of calves (C1–C10) originated from a single dairy farm in Southeastern Pennsylvania and were enrolled in the study at 16–20 days old (approximately 45kg birth weight). The calves received acidified milk until eight weeks of age, after which they gradually transitioned to a diet comprising grain and hay until approximately 9 to 12 weeks old (S1 Table). Grain consumption was provided, with quantities adjusted based on the animals' age. Throughout the study, fecal samples were collected from each calf (n = 10) at six specific time points: weeks 2, 4, 6, 8, 10, and 12 (referred to as Wk2, Wk4, Wk6, Wk8, Wk10, and Wk12). Fecal samples were collected 2h post-feeding and were obtained via rectal stimulation, placed into plastic deli containers, and transported on ice to the laboratory, where they were stored at −20°C until DNA extraction.

## DNA Extraction, amplification, and sequencing

Genomic DNA was extracted from fecal samples (approximately 250 mg) using the repeated bead beating and column (RBB + C) method, followed by purification with the QIAmp Fast DNA Stool Mini Kit (Qiagen Sciences, Germantown, MD), as detailed by Yu and Morrison [11]. Subsequently, the V6-V8 region of the archaeal 16S rRNA gene was PCR-amplified in triplicate using specific primers i958aF (5′–AATTGGAKTCAACGCCKGR–3′) and i1378aR (5′–TGTGTGCAAGGAGCAGGGAC–3′). Both sets of primers were barcoded with Golay codes for multiplexing. Polymerase chain reaction (PCR) conditions involved an initial denaturing step, followed by 20 cycles, and a final extension step. Amplicons from each DNA sample were pooled, purified, and subjected to sequencing on the Illumina MiSeq platform at the PennCHOP Microbiome Core, University of Pennsylvania. The rt-PCR assay on rumen samples was performed as described in Pitta et al. [12].

## Bioinformatics and statistical analysis

The archaeal amplicon sequences were processed through the QIIME2 version 2020.6 pipeline [13]. The raw sequences were de-multiplexed and assigned to amplicon sequence variants (ASV) using the DADA2 plugin [14]. Multiple sequence alignment was carried out with MAFFT [15] and sequences were filtered to remove highly variable positions using default settings. FastTree 2 [16] was used to construct and root a phylogenetic tree using default settings. Taxonomic classification was conducted using a pre-trained Naive Bayes classifier trained on the Greengenes version 13.8 database [17] for the 16S rRNA region spanning the V4-V6 region. Weighted and unweighted UniFrac beta diversity metrics and Observed species and Shannon diversity alpha diversity metrics were estimated using q2- core-metrics-phylogenetic after samples were rarefied to 4,674 reads per sample.

To analyze the alpha diversity metrics and the distribution of archaeal genera, we employed generalized estimating equations (GEE) with the geeglm function from the geepack R package. For the evaluation of beta diversity matrices, we utilized a nonparametric permutational multivariate ANOVA (PERMANOVA) test, which was implemented using the vegan package in R. To facilitate comparisons, the raw read counts from the 16S rRNA ASV abundance table were collapsed at the genus rank and compositionally normalized to relative abundance, ensuring each sample summed to one and subsequently log-transformed.

## Sequence availability

The fecal archaeal 16s rRNA sequences have been deposited in the NCBI database under BioProject accession number PRJNA1046510.

## Results

### Sequencing details

A total of 685,830 raw reads were generated from 56 samples, yielding a mean of 12,247 (±4011) reads per sample. Four experimental samples were removed from the sequencing library for containing too few reads (< 500), and a DNA blank sample containing <10 reads was also removed from the analysis. After quality filtering and denoising, we identified 38 unique Amplicon Sequence Variants (ASVs) from about 564,524 sequencing variants. Notably, this diversity was relatively low compared to bacterial studies of the same samples. The ASVs ranged between 16,522 to 4,674 sequences per sample. To enable unbiased comparisons between samples with different sequencing depths, reads were rarefied at the minimum sequencing depth (4,674 reads) for diversity analysis.

### Intra-sample diversity (alpha diversity)

To investigate temporal patterns in calf archaeal communities, we evaluated species richness (quantifying the number of unique taxa; Fig 1A) and the Shannon Diversity Index (considering both species richness and relative abundances; Fig 1B). Species richness did not reveal significant differences between time points (GEE; *P* = 0.41). However, the Shannon diversity index indicated lower archaeal diversity until Wk6, corresponding to the period when calves received the milk only diet. Subsequent increases in grain consumption were associated with

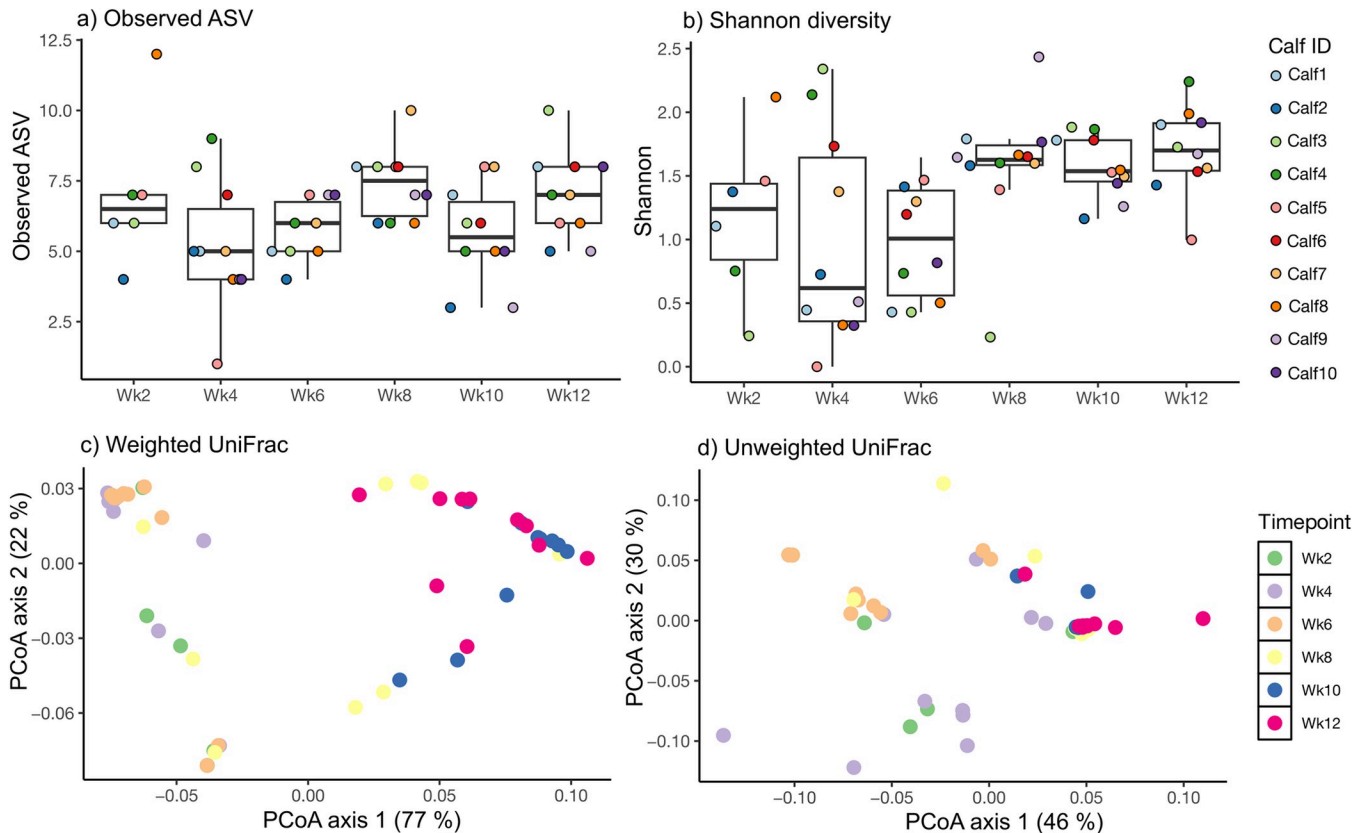

**Fig 1. Within-sample diversity (alpha diversity) and between sample diversity (beta diversity).** Fecal archaeal diversity in a cohort of 10 calves across six sampling timepoints. (a) Species richness, (b) Shannon diversity. Principal Coordinate Analysis (PCoA) illustrating the fecal archaeal communities based on (c) Weighted UniFrac distances and (d) Unweighted UniFrac distances.

elevated methanogen diversity from Wk10 through Wk12. Pairwise comparisons revealed statistically significant increases in Shannon diversity for Wk10 and Wk12 compared to Wk6 (GEE; $P < 0.001$).

### Inter-sample diversity (between diversity)

We further investigated overall differences in archaeal DNA community composition using phylogeny-based weighted and unweighted UniFrac metrics (Fig 1C and 1D), employing principal coordinates analysis (PCoA) and permutational multivariate ANOVA (PERMANOVA) pairwise analyses. Our findings showed significant ($P = 0.001$) age-related differences in archaeal communities based on both weighted (reflecting relative abundances of commonly present bacterial taxa) and unweighted (indicating presence-absence information of archaeal taxa) distance matrices. Specifically, samples from Wk2 to Wk6 exhibited greater similarity to each other and were distinct from the cluster that formed from Wk8 through Wk12.

### Taxonomical composition of fecal archaeal communities

Overall, the most dominant archaeal populations were identified as *Methanobrevibacter* (up to 63%), followed by *Methanosphaera*, which accounted for up to 37% of total archaeal abundance. *Methanobrevibacter* was the most prevalent genus at Wk2, Wk4, and Wk6 (96.2%), but its relative abundance decreased at Wk8, Wk10, and Wk12 (33.8%). In contrast, *Methanosphaera* increased gradually with time and was most abundant at Wk8, Wk10, and Wk12 (66.2%) (Fig 2A). Subsequently, our investigation delved into the quantification of copy numbers for two specific species, namely *Methanobrevibacter ruminantium* (*M. ruminantium*) and *Methanosphaera stadtmanae* (*M. stadtmanae*), along with the 16S rRNA gene, which is universally present in all methanogenic, employing RT-qPCR analysis (Fig 2B). The combined actual

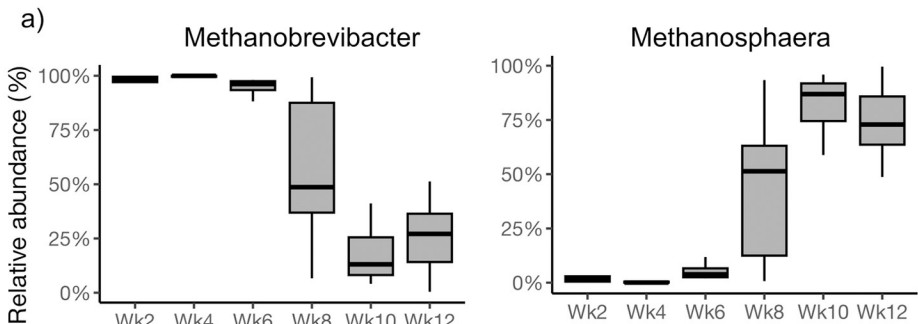

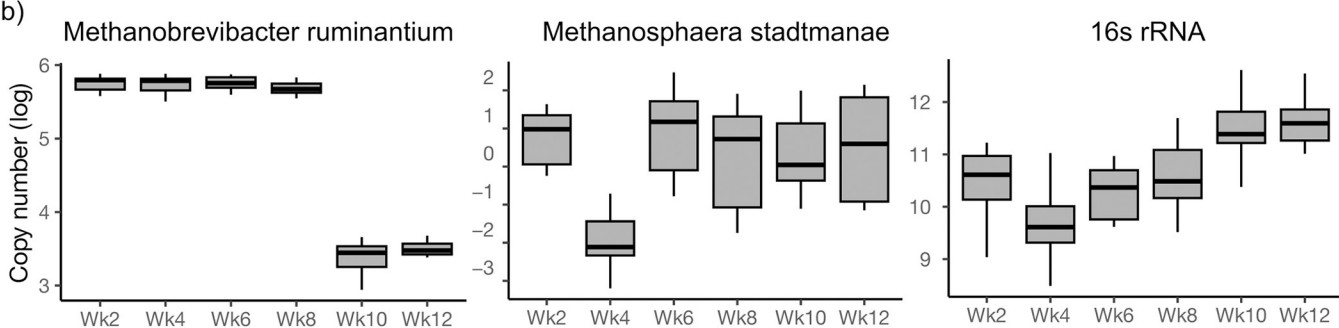

**Fig 2. Fecal archaeal composition in a cohort of 10 calves across six sampling timepoints.** (a) comparison of archaeal composition at genus level based on 16s rRNA amplicon sequencing analysis (b) comparison of archaeal composition at species levels based on RT-qPCR analysis.

copy number of *M. ruminantium* and *M. stadtmanae* was found to be $10^{11}$ cells per ml of fecal contents, slightly lower than the adult cow count of $10^{12}$ per ml. Specifically, *M. ruminantium* accounted for $10^{5-6}$ cells per ml, whereas *M. stadtmanae* contributed approximately $10^{1-2}$ cells per ml in calf feces. For statistical analysis and visual representation, the actual copy numbers were transformed into log copy numbers.

The log copy number of the 16S rRNA gene exhibited a significant increase at Wk10 and Wk12 compared to Wk2 (GEE; $P < 0.001$). Notably, the copy number of *M. ruminantium* began to decline at Wk8 (GEE; $P < 0.001$), while *M. stadtmanae* displayed the opposite trend, albeit with marginal significance (GEE; $P = 0.06$). Furthermore, the RT-qPCR analysis agreed with the findings from 16S rRNA amplicon sequencing. *Methanobrevibacter* emerged as the most dominant genus at Wk2, Wk4, and Wk6, diminishing thereafter from Wk8. In contrast, *Methanosphaera*, while less dominant at Wk2 and Wk4, gradually increased in prevalence from Wk6 to Wk12.

### The interplay between predominant bacteria and methanogens

To further our understanding of bacteria that are associated with the two methanogens, we computed a Spearman correlation matrix between predominant bacteria ($> 0.01\%$; [10]) and two methanogenic genera (Fig 3). It was obvious that *Methanobrevibacter* and *Methanosphaera* demonstrated a very strong negative correlation with each other. Of the 52 predominant bacterial genera included in this analysis, approximately 30 bacterial genera showed either positive or negative correlations with the two methanogenic genera. Consequently, of the 30 bacterial genera that showed positive correlations with *Methanobrevibacter* were negatively correlated with *Methanosphaera* and vice versa. Notably, only 8 out of 30 genera showed positive correlations with *Methanobrevibacter* whereas 22 of 30 bacterial genera showed positive correlations with *Methanosphaera*. The genera that showed positive correlations with *Methanobrevibacter* were *Collinsella*, *Bacteroides*, *Parabacteroides*, *Blautia*, *Clostridia*, *Eubacterium*, *Faecalibacterium* and *Streptococcus*, with the *Faecalibacterium* showing the strongest correlation with *Methanobrevibacter*. While these genera were negatively correlated with *Methanosphaera*, some of the well-known bacterial genera such as *Treponema*, *Sharpea*, *Mogibacteriaceae* and *S24-7* were among the strongly positively correlated with *Methanosphaera*.

### Discussion

Enteric methane is exclusively produced by methanogenic archaea (a special group of microbes that exist in synergy with other fermenting microbes) during microbial fermentation of feeds in the rumen. While it is known that methanogens are important for the adult rumen to keep the $H_2$ concentrations low for fermentation reactions to occur, information on their colonization, population dynamics and the evolution of their interactions with other microbes during the preweaning stage, and as they transition to mature ruminants is not very well explored. The goal of this study is to investigate the dynamics in methanogen communities in preweaned calves from birth through weaning transition.

Previously, we [10] investigated the bacterial temporal dynamics and its progression in calves from birth through 12 weeks of age (weaning transition) and reported there is a considerable variation in bacterial communities during the preweaning period and stabilized at weaning. This study is an accompaniment to Hennessy et al., [10] where we investigated methanogen communities from birth through weaning at Wk2, Wk4, Wk6, Wk8, Wk10, and Wk12 in a cohort of 10 calves that were fed acidified milk. Similar to the bacterial changes, we observed a considerable fluctuation in the methanogenic communities, suggesting that microbial communities were dynamic, as revealed by Observed Species metrics, and appeared to

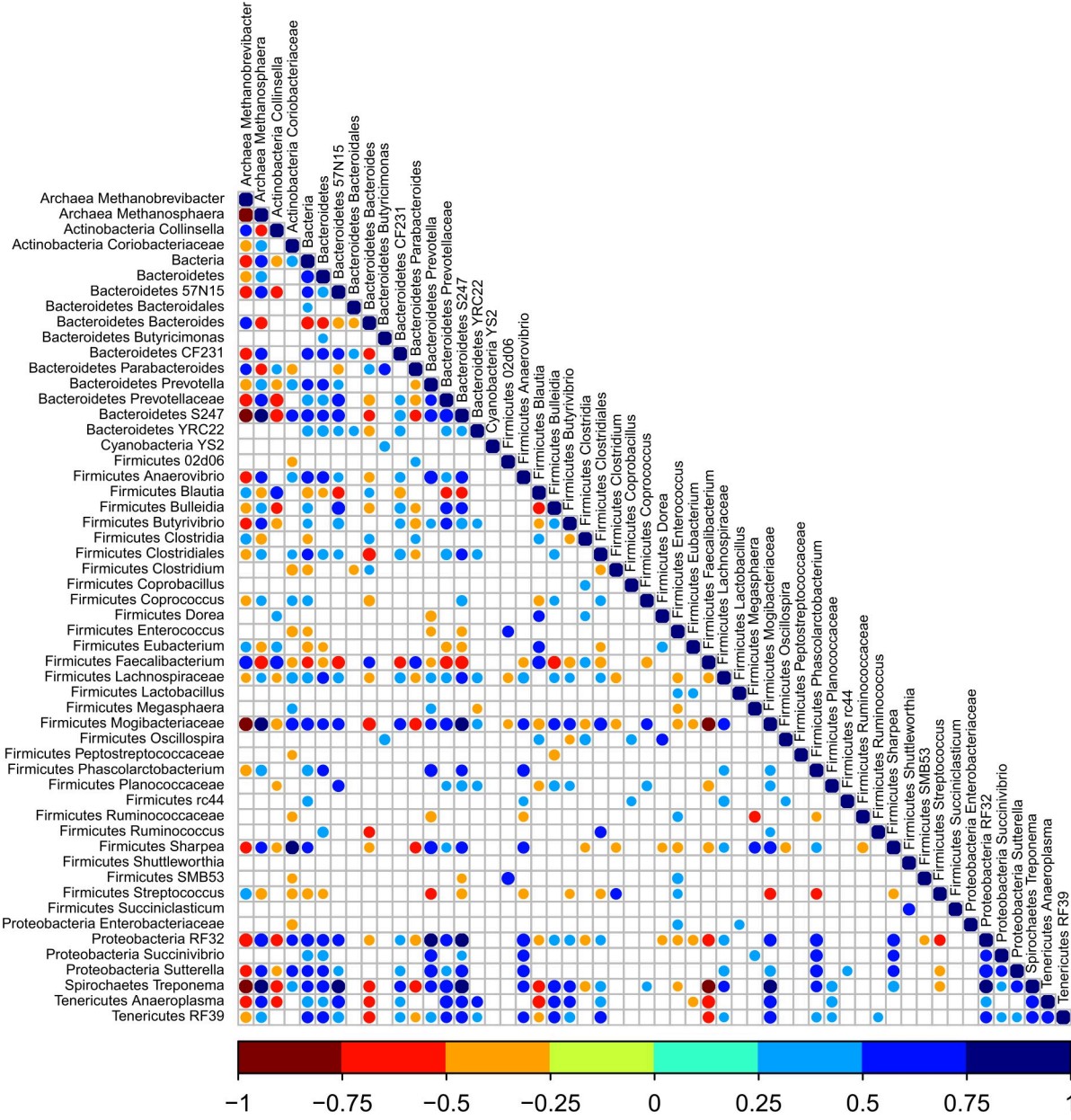

**Fig 3. Analysis of association patterns between bacterial and archaeal microbial lineages scored using Spearman correlation.** Individual taxa were considered present in a sample if their sequence proportion was at least 0.01% of relative abundance. Correlations are shown by the color code (blue: positive correlations, red: negative correlations).

follow the changes in diet-induced bacterial communities. Notably, the Shannon diversity index and the unweighted and weighted methanogenic communities show the formation of two major clusters (grouping patterns), with one cluster from Wk2 through Wk6 and the other from Wk8 through Wk12. Congruently, the first cluster (Wk2 through Wk6) was dominated by genus *Methanobrevibacter* and the other cluster was dominated by genus *Methanosphaera*. The 16S rDNA sequencing results also showed that *Methanobrevibacter* was dominant at Wk2 through Wk6 and then gradually declined from Wk8 through Wk12 whereas *Methanosphaera* was minimal until Wk6 gradually increased from Wk8 through

Wk12. Surprisingly, at any given timepoint, either *Methanosphaera* or *Methanobrevibacter* existed but did not co-occur according to the correlation analysis. This means that both these genera compete with each other and have been reported to be negatively correlated with each other [18]. Lineages of *Methanobrevibacter* require $CO_2$ and $H_2$, whereas lineages of *Methanosphaera* require methanol and $H_2$. Irrespective of different substrates ($CO_2$ and methanol), they both compete for $H_2$. Interestingly, $H_2$ thresholds for *Methanosphaera* are much lower than *Methanobrevibacter* [19], which means that as long as methanol is available as a substrate, *Methanosphaera* will dominate and has the potential to outcompete *Methanobrevibacter*. It is possible to speculate that $CO_2$ may be a dominant substrate when calves are entirely on milk (until Wk6), whereas methanol becomes available as calves begin to eat grain. An increase in methanol favors the growth of *Methanosphaera* from Wk6, thus replacing *Methanobrevibacter* for the remainder of the study period (until Wk8).

Although the 16S rDNA data provides information on relative abundance, this approach is limited for not providing quantitative information. Our RT-qPCR data aligns well with the 16S rRNA results showing variability in the first six weeks and then steadily increases in copy number from wk6 through Wk12. The time period (from Wk2 through Wk4) when *Methanobrevibacter* was dominant, the 16S rDNA copy number as analyzed by RT-qPCR data was lower ($10^{9-11}$) whereas the 16S rDNA copy number steadily increased when *Methanosphaera* increased from Wk6 through Wk12 ($10^{12}$ copies per 100ng of DNA). We further analyzed the copy number of *M. ruminantium* throughout the study, and results showed that it was only $10^{5-6}$ until Wk8, whereas the copy number of *M. stadtmanae* had $10^2$ copies per 100ng of DNA throughout the study period except at Wk4. It was interesting to note that the absolute count of unique ASVs from 16s rRNA amplicon sequencing analysis was in the order of 7,000 at Wk2 through to 12.000 at Wk5 and Wk6, which agrees with the increasing trend of methanogen 16S rDNA copy number as analyzed by RT-qPCR. The number of 16SrDNA reads assigned to *Methanobrevibacter* averaged 8,000 until Wk6 and then gradually decreased. In contrast, the numbers of 16S rDNA reads that were assigned to *Methanosphaera* substantially increased (S2 Table) from Wk6 and peaked at Wks 10 and 12. It is interesting to note the congruency between rt-PCR and 16S rDNA raw counts assigned to either *Methanobrevibacter* or *Methanosphaera* genus, indicating these two genera make up the dominant early-life methanogen communities. These findings also open the opportunity to develop novel mitigation strategies to inhibit either one or both genera, thus effectively curbing methane emissions later in life.

Previously, we reported that certain bacteria form specific associations with certain methanogens forming microbial networks in the rumen of adult dairy cattle [18]. As we observed fluctuations in both bacteria and methanogens, we hypothesized that these specific networks may form very early in life, especially with *Methanobrevibacter* dominant until 6 weeks and *Methanosphaera* later in the preweaning period. Calves were fed acidified milk during the first few weeks of life with *ad libitum* access to grain. In modern dairy systems, dairy calves are fed differently processed milk, such as pasteurized milk, acidified milk, and waste milk, during the first few weeks of life. Of these, feeding acidified milk has been practiced on farms to reduce the pathogen burden and increase the shelf life of dairy products [20]. Acids such as formic acid and citric acid are used to acidify milk to a range of pH between 4.0 to 4.5. Studies have shown that feeding acidified milk to dairy calves resulted in improved weight gains and reduced calf scours [21, 22]. Furthermore, our previous study [23] also showed that feeding acidified colostrum, followed by acidified milk, led to an increase in the proliferation of essential bacteria such as *Faecalibacterium* with probiotic-like properties. These data indicated that feeding acidified milk can be considered one of the feeding practices in the management of neonatal dairy calves.

We did report in our previous paper [10] that the consumption of milk was reduced and compensated with an increase in grain around Wk6. However, the calves were weaned from milk and were switched to grain completely by Wk8. This transition to a solid diet explains a different bacterial profile, and enriched with methanol-utilizing *Methanosphaera* population. The correlation analysis between bacteria and methanogens in calves resembled that of adults suggesting the formation of these associations is established very early in life and continues to persist through adulthood. However, *Methanobrevibacter* showed positive associations with *Collinsella*, *Faecalibacterium*, *Blautia*, *Bacteroides* and *Parabacteroides* which are very critical for early phase of calf's life. In contrast, *Methanosphaera* have positive associations with many bacteria including some important bacteria such as *Prevotella*, *S24-7*, *Sharpea*, *Bulledia*, *Mogibacteriaceae* and *Butyrivibrio*. Particularly, the interactions of *Methanosphaera* and specific bacteria such as *Prevotella*, *Bulledia*, *Lachnospiraceae* and *Mogibacteriaceae* identified in calf feces from this study were similar to those reported in the mature rumen of dairy cattle as reported in Kaplan-Shabtai et al. [18]. The associations of *Methanosphera* and specific bacteria such as *Prevotella*, *Bulledia*, *Butyrivibrio*, *Lachnospiraceae*, *Mogibacteriaceae*, *Succinivibrionaceae* and *Sharpea* were also characteristic of the rumen in cows with low-methane yield phenotype as reported in [19]. Some genera, such as *Prevotella*, *Bulledia*, *Lachnospiraceae*, *Sharpea*, and *Mogibacteriaceae*, as reported in Kaplan-Shabtai et al., [18] and Stepanchenko et al., [19], have a positive association with *Methanosphaera*, may have been established during the preweaning period, as noted in this study.

## Conclusion

Collectively, these data indicate that *Methanobrevibacter* dominates in the rumen when dairy calves receive milk products in the early weeks of life and form associations with bacteria that are critical players during the preweaning period. However, with the introduction of the high grain diet, these *Methanobrevibacter*-bacteria interactions are outcompeted by *Methanosphaera* driven $H_2$ transactions and replaced with their corresponding bacterial partners. These findings imply that bacteria-methanogens associated are possibly imprinted in the early phase of life with long lasting patterns that are carried into adulthood. Further studies investigating the effects of different mitigation strategies at different phases of preweaning period will be needed to interfere with methanogen colonization during preweaning period to curb methane emissions later more effectively in the life of ruminants.

## Supporting information

**S1 Table. Amount of acidified milk and starter grain given per calf per week and composition of starter grain.**
(DOCX)

**S2 Table. Sequencing data and the predominant methanogenic archaea identified in individual samples.**
(CSV)

## Acknowledgments

The authors would like to thank the PennCHOP Sequencing Core for their assistance with this project. We would also like to thank Walmoore Holsteins, Inc. and Lindsay Stover for allowing us to conduct this study at their facility.

## Author Contributions

**Conceptualization:** Dipti Pitta.

**Data curation:** Nagaraju Indugu, Dipti Pitta.

**Formal analysis:** Nagaraju Indugu.

**Funding acquisition:** Dipti Pitta.

**Investigation:** Nagaraju Indugu, Dipti Pitta.

**Methodology:** Nagaraju Indugu, Kapil S. Narayan, Meagan L. Hennessy.

**Project administration:** Dipti Pitta.

**Resources:** Dipti Pitta.

**Software:** Nagaraju Indugu.

**Supervision:** Dipti Pitta.

**Validation:** Nagaraju Indugu.

**Visualization:** Nagaraju Indugu.

**Writing – original draft:** Nagaraju Indugu, Dipti Pitta.

**Writing – review & editing:** Nagaraju Indugu, Kapil S. Narayan, Dipti Pitta.

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
