## [Decision Letter · Decision Letter 0]

10 Jul 2024

PONE-D-24-07201Establishment of methanogen bacterial interactions during the preweaning period of dairy cattlePLOS ONE

Dear Dr. Pitta,

Thank you for submitting your manuscript to PLOS ONE. After careful consideration, we feel that it has merit but does not fully meet PLOS ONE’s publication criteria as it currently stands. Therefore, we invite you to submit a revised version of the manuscript that addresses the points raised during the review process.

We look forward to receiving your revised manuscript.

Kind regards,

Ibukun M. Ogunade, Ph.D.

Academic Editor

PLOS ONE

“This research was funded by the United States Department of Agriculture National Institute of Food and Agriculture (Grant #PENV570262; https://nifa.usda.gov/us-states-and-territories/pennsylvania).”

Reviewers' comments:

Reviewer's Responses to Questions

**Comments to the Author**

1. Is the manuscript technically sound, and do the data support the conclusions?

Reviewer #1: Yes

Reviewer #2: Yes

2. Has the statistical analysis been performed appropriately and rigorously? 

Reviewer #1: Yes

Reviewer #2: Yes

3. Have the authors made all data underlying the findings in their manuscript fully available?

Reviewer #1: Yes

Reviewer #2: Yes

4. Is the manuscript presented in an intelligible fashion and written in standard English?

Reviewer #1: Yes

Reviewer #2: Yes

5. Review Comments to the Author

Reviewer #1: Overall, this is a beautiful study revealing microbial dynamics in developing animals during the preweaning period of dairy cattle. This is crucial for understanding and optimizing rumen development, which significantly impacts the future productivity and health of the animals.

Comments

Line 84-88: Please rewrite the objective and your hypothesis to succinctly conform with the abstract. There is a disjoint in the sentence especially line 88……. which further dictates….. Sentences like this should not stand here.

Line 98: Remove ad libtum from this sentence and leave as…… grain consumption was provided………

Line 99: How did you collect your fecal samples? Was the feces collected via stimulation or was picked from the floor……. It would be important to have a line of sentence describing this.

Line 220: Replace accompaniment with follow up study…….

Line 144: Rarefied or rarified… Typo error, this should be Rarefied.

Reviewer #2: The manuscript presents a valuable contribution to understanding methanogen colonization in neonatal dairy calves and its potential impact on methane emissions. The study is methodologically sound and well-referenced, providing a comprehensive analysis of the temporal dynamics of methanogenic archaea and their interactions with bacteria.

However, some information about Materials and methods are missing, likewise some improvements need to be made.

Abstract:

Include age and average weight of the calves at the point of the study

Introduction:

Add references at line 56, and 58.

All acronyms should be defined.

While some of the target audiences are likely familiar with technical terms, ensuring that complex terminology (e.g., "methylotrophic methanogens", "hydrogenotrophic methanogens") is clearly defined upon first use can enhance accessibility.

The knowledge gap mentioned in lines 81-83 is critical but could be more explicitly stated. A clear statement of what is known versus what is not would sharpen the focus.

Materials and Methods:

Include the average weight of the calves at the point of the study

Why wasn’t a control group (group of calves not receiving the same diet) included in the study to help provide comparative insights?

Provide detailed information about the acidified milk and the grain and hay diet content (Diet Table).

Why were fecal samples collected only and rumen fluid samples were not collected to give more information about the ruminal content and the microbial populations of the calves, similar to the study by Friedman et al., 2017.

Details of fecal collection procedure should be included

What time during the day were the fecal samples collected from each calf, was it after or before meals?

Results

Line 138: Were these four experimental samples removed from other analyses?

Line 179 needs to be improved (Methanogenic)

Line 226 shows should be show

Were residuals checked for normality, and how were potential outliers handled?

Discussion

The transition between discussing bacterial and methanogen communities seems a bit abrupt, improving the flow would help readability

Explain why acidified milk was chosen and how it might influence the microbial dynamics compared to other feeding strategies

Rephrase line 232 to “Surprisingly, the correlation analysis revealed that, at any given time point, only one of the two genera was present, and they did not co-occur.” to improve clarity.

Simplifying the language where possible and defining key terms would make the discussion more accessible.

The use of terms like "flux" and "clusters" needs to be clearly defined in the context of microbial community dynamics to avoid ambiguity.

Lines 263-266: The mention of similar interactions in mature rumen and low-methane yield phenotype cows is interesting but needs more context and explanation. How do these findings compare with other studies on methanogen-bacteria interactions?

Create a sub-heading for the Conclusion

6. PLOS authors have the option to publish the peer review history of their article (what does this mean?). If published, this will include your full peer review and any attached files.

Reviewer #1: **Yes: **Godstime Taiwo

Reviewer #2: **Yes: **Modoluwamu Idowu

---

## [Author Response · Author response to Decision Letter 0]

3 Sep 2024

Editors’ comments

AU: We have revised the manuscript to adhere to PLOS ONE's style requirements. This includes updates to figure citations, file naming conventions, and references to align with the guidelines provided.

“This research was funded by the United States Department of Agriculture National Institute of Food and Agriculture (Grant #PENV570262; https://nifa.usda.gov/us-states-and-territories/pennsylvania).” 

AU: As requested, we have included the Role of Funder statement in the cover letter. 

AU: We have updated our manuscript according to PLOS submission guidelines.

AU: We have reviewed and updated references and citations

5. Review Comments to the Author

Reviewer #1: Overall, this is a beautiful study revealing microbial dynamics in developing animals during the preweaning period of dairy cattle. This is crucial for understanding and optimizing rumen development, which significantly impacts the future productivity and health of the animals.

Thank you for your comments. We have sought to address the specific critiques below.

Comments

Line 84-88: Please rewrite the objective and your hypothesis to succinctly conform with the abstract. There is a disjoint in the sentence especially line 88……. which further dictates….. Sentences like this should not stand here.

AU: We have revised the objective and hypothesis for clarity. The sentence has been improved as suggested (Lines 84-94).

Line 98: Remove ad libtum from this sentence and leave as…… grain consumption was provided………

AU: We have removed the term "ad libitum" as suggested.

Line 99: How did you collect your fecal samples? Was the feces collected via stimulation or was picked from the floor……. It would be important to have a line of sentence describing this.

AU: We collected fecal samples via rectal stimulation. This information has been added to the manuscript (Lines 107-110).

Line 220: Replace accompaniment with follow up study…….

AU: Agreed and corrected. 

Line 144: Rarefied or rarified… Typo error, this should be Rarefied.

AU: Thank you. We have corrected it to "Rarefied."

Reviewer #2: The manuscript presents a valuable contribution to understanding methanogen colonization in neonatal dairy calves and its potential impact on methane emissions. The study is methodologically sound and well-referenced, providing a comprehensive analysis of the temporal dynamics of methanogenic archaea and their interactions with bacteria. However, some information about Materials and methods are missing, likewise some improvements need to be made.

Thank you for your comments. We have sought to address the specific critiques below.

Abstract:

Include age and average weight of the calves at the point of the study

AU: The age of the calves and average birth weight is included in the abstract as suggested (Lines 32-35).

Introduction:

Add references at line 56, and 58.

AU: We have revised the sentences as requested and added two references: Johnson & Johnson (1995) and Wolin (1981).

All acronyms should be defined.

While some of the target audiences are likely familiar with technical terms, ensuring that complex terminology (e.g., "methylotrophic methanogens", "hydrogenotrophic methanogens") is clearly defined upon first use can enhance accessibility.

AU: We have defined both "methylotrophic methanogens", and "hydrogenotrophic methanogens (Lines 74-77)

The knowledge gap mentioned in lines 81-83 is critical but could be more explicitly stated. A clear statement of what is known versus what is not would sharpen the focus.

AU: We have revised the text to clearly state the knowledge gap (Lines 84-88).

Materials and Methods:

Include the average weight of the calves at the point of the study

AU: We included the average birth weight of the calves (lines 102-103).

Why wasn’t a control group (group of calves not receiving the same diet) included in the study to help provide comparative insights?

AU: The primary objective of this study was to understand the temporal patterns of methanogenic archaea, rather than to compare different diets. As such, a control group was not included. All calves followed the same dietary protocol: they received acidified milk until 8 weeks of age and were then switched to a diet of grain and hay, as per the standard feeding schedule on our dairy farm. As such there is no treatment but we compared methanogenic communities by age of calves.

Provide detailed information about the acidified milk and the grain and hay diet content (Diet Table).

AU: We have now included a detailed Diet Table as Supplementary Table 1 in the manuscript, which provides information on the composition of the acidified milk, grain, and hay diets.

Why were fecal samples collected only and rumen fluid samples were not collected to give more information about the ruminal content and the microbial populations of the calves, similar to the study by Friedman et al., 2017.

AU: Rumen is not fully developed, and calves are considered monogastric until weaning. According to Friedman et al paper, samples were collected at Wk2 and then 2 months which is not similar to our study. Our goal was to follow temporal dynamics and sampling for rumen contents at that frequency will be considered invasive sampling. Collecting rumen at birth through 4-6 weeks of age is also considered invasive and may not represent true composition of rumen microbiota as rumen is still developing. Therefore, we relied on fecal sampling.

Details of fecal collection procedure should be included

AU: The fecal collection procedure is now described in (lines 107-110).

What time during the day were the fecal samples collected from each calf, was it after or before meals?

AU: We typically collect fecal samples 2h post-feeding, this information has been updated in text (Line 108).

Results

Line 138: Were these four experimental samples removed from other analyses?

AU: Yes, we removed the four experimental samples from all microbiome analyses.

Line 179 needs to be improved (Methanogenic)

AU: We have replaced “Methanogenic” with “Archaea”.

Line 226 shows should be show

AU: We have replaced “shows” with “show”.

Were residuals checked for normality, and how were potential outliers handled?

AU:

We used generalized estimating equations (GEE) statistical model to analyze alpha diversity and for taxonomy. In GEE, the primary focus is on the population-averaged effects rather than individual-level predictions, and as such, the normality of residuals is not a strict requirement. GEE is robust to misspecifications of the working correlation structure, and the estimates of the regression coefficients are consistent even if the residuals are not normally distributed. Therefore, we did not check residuals for normality, as it is not a prerequisite for the validity of GEE results. 

In our analysis, we used boxplots and interquartile range (IQR) statistics to identify potential outliers and found two animals at Week 2 that were outliers. We then ran the GEE model without these two observations. The results were consistent whether or not we included the outliers, indicating that they did not significantly impact our findings.

Discussion

The transition between discussing bacterial and methanogen communities seems a bit abrupt, improving the flow would help readability

AU: Thank you, we have revised the sentence as suggested (Lines 268-281). 

Explain why acidified milk was chosen and how it might influence the microbial dynamics compared to other feeding strategies

AU: We have added justification for using acidified milk (Lines 282-300).

Rephrase line 232 to “Surprisingly, the correlation analysis revealed that, at any given time point, only one of the two genera was present, and they did not co-occur.” to improve clarity.

AU: We have rephrased the sentence as suggested (Lines 249-264). 

Simplifying the language where possible and defining key terms would make the discussion more accessible.

The use of terms like "flux" and "clusters" needs to be clearly defined in the context of microbial community dynamics to avoid ambiguity.

AU: flux eliminated, grouping patterns added in parenthesis for clusters

Lines 263-266: The mention of similar interactions in mature rumen and low-methane yield phenotype cows is interesting but needs more context and explanation. How do these findings compare with other studies on methanogen-bacteria interactions?

AU: We have revised the sentences as suggested (Lines 325-333)

Create a sub-heading for the Conclusion

AU: A separate sub-heading has been created for the Conclusion.

---

## [Editor Report · Decision Letter 1]

5 Sep 2024

Establishment of methanogen bacterial interactions during the preweaning period of dairy cattle

PONE-D-24-07201R1

Dear Dr. Pitta,

We’re pleased to inform you that your manuscript has been judged scientifically suitable for publication and will be formally accepted for publication once it meets all outstanding technical requirements.

Kind regards,

Ibukun M. Ogunade, Ph.D.

Academic Editor

PLOS ONE
---

## [Editor Report · Acceptance letter]

11 Sep 2024

PONE-D-24-07201R1 

PLOS ONE

Dear Dr. Pitta, 

I'm pleased to inform you that your manuscript has been deemed suitable for publication in PLOS ONE. Congratulations! Your manuscript is now being handed over to our production team.

Kind regards, 

on behalf of

Dr. Ibukun M. Ogunade 

Academic Editor

PLOS ONE